# Does Small-Scale Livestock Production Use a High Technological Level to Survive? Evidence from Dairy Production in Northeast-ern Michoacán, Mexico

**DOI:** 10.3390/ani11092546

**Published:** 2021-08-30

**Authors:** Luis Manuel Chávez-Pérez, Ramón Soriano-Robles, Valentín Efrén Espinosa-Ortiz, Mauricio Miguel-Estrada, María Camila Rendón-Rendón, Randy Alexis Jiménez-Jiménez

**Affiliations:** 1Doctorado en Ciencias Agropecuarias, Universidad Autónoma Metropolitana, Unidad Xochimilco, Calzada del Hueso 1100, Ciudad de México 04960, Mexico; 2Departamento de Economía, Administración y Desarrollo Rural, Facultad de Medicina Veterinaria y Zootecnia, Universidad Nacional Autónoma de México, Avenida Universidad 3000, Ciudad de México 04510, Mexico; veoee1@hotmail.com (V.E.E.-O.); mauriciome@fmvz.unam.mx (M.M.-E.); 3Laboratorio de Recursos Socioambientales y Sustentabilidad, Departamento de Biología de la Reproducción, Área de Investigación en Reproducción Animal Asistida, Universidad Autónoma Metropolitana, Unidad Iztapalapa, San Rafael Atlixco 186, Ciudad de México 09340, Mexico; ramon@xanum.uam.mx; 4Instituto de Ciencias Agropecuarias y Rurales (ICAR), Universidad Autónoma del Estado de México, Toluca 50295, Mexico; mcrendon@gmail.com

**Keywords:** multivariate analysis, Holstein model, small-scale dairy systems, family labour

## Abstract

**Simple Summary:**

Small-scale dairy production is a activity in rural areas, as it generates daily income and contributes to food security. In Mexico, from the 1980s, economic policies were promoted that led to the modernisation and concentration of milk production, which caused many small-scale production units to disappear and reduce their contribution to the national supply. Some researchers report that the production units that survived were those that specialised and incorporated a high level of technology. The objective of this work was to identify the technological level and socioeconomic conditions of dairy production units in the northeastern region of Michoacán, Mexico, in order to know the strategies that have allowed them to survive. Through statistical analysis, four groups of farms with different levels of technology were identified. The clusters that predominate in this region use a low level of technology and have low productivity, finding their strengths in the diversification of activities they carry out and the use of family labour. Public policies should be directed, for each cluster, in a differentiated manner, prioritising the strengthening of the aspects that have kept them going in the present, rather than the incorporation of a high technological level in their production units.

**Abstract:**

The objective of this study was to identify the technological level used by dairy farmers in the northeastern region of Michoacán, Mexico, through a characterisation of small-scale dairy production units, as well as to learn about the socioeconomic conditions that have enabled them to survive in the current context. A semi-structured interview was applied to 114 production units, chosen by stratified random sampling. The interview included technological, production and socioeconomic aspects. Twenty-eight variables were initially explored and 12 were used for multivariate analysis, which included Principal Component Analysis, Hierarchical Cluster Analysis and K-means Cluster. The characterisation carried out showed that the production units that predominate in northeastern Michoacán have survived with a low technological level, having as strengths the diversification of their activities and the use of family labour. On the contrary, production units with a high technological level and high productivity are few and less diversified. This shows the need to generate differentiated public policies for each cluster, aimed at strengthening the aspects that have allowed them to survive and guaranteeing a market for their production, before promoting the use of technologies.

## 1. Introduction

Dairy production provides the main income for many farmers worldwide [1], in addition to contributing to the livelihoods, food security and nutrition of approximately 150 million households [2]. In developing countries, it is estimated that more than 80% of dairy production is carried out in Small-Scale Dairy Production Units (SSDPU) [2]. In Mexico, these represent 78%, equivalent to 142,167 SSDPU [3]. Several authors mention that these SSDPU generate a constant economic income for producers [4], improve people’s diets [5,6], use family resources such as labour, land, water and capital [7], adapt to diverse physical spaces and environments [8], generate employment in rural areas and add value to forage and agricultural by-products [9].

However, the contribution of SSDPU to the national milk supply has been reduced since the 1980s, with the promotion of economic policies aimed at subsidising consumption, price control and imports of milk powder at low prices [10]. Since those years, intensive systems have increased their share of the national supply from 24% (1617.84 million litres of milk) to 51% (5462.93 million litres of milk), while SSDPU reduced their contribution from 21% (1415.61 million litres) to 9% (964.05 million litres) [11,12]. This decrease in the contribution of the SSDPU was mainly due to the fact that the intensive systems increased their production and productivity and thus its contribution to the national supply. Likewise, it was due to the decrease in the number of SSDPU in some regions of the country—for instance, from 1994 to 2004, between 25% and 41% of the producers in the highlands of Jalisco, abandoned the activity [13], or as in Chipilo, Puebla, where 33% also abandoned the milk production [14]—for the few technological changes that they incorporated [14], this resulted in low increases in productivity and a decrease in their contribution to supply national milk. Because of this, a trend towards the modernisation, concentration and centralisation of dairy production in the country has been observed [12,14,15].

This trend has not been exclusive to Mexico. This is very noticeable in developed countries in Europe, Oceania and in the United States [16]; for example, in Sweden, the dairy industry reduced the number of farms by 40% between 2001 and 2007, increasing the production and competitiveness of the remaining farms [17]. Meanwhile, in Wisconsin, USA, in order to survive, small family farmers found it necessary to make investments to modernise their dairy facilities [18]. In developing countries, the information is not extensive but these trends are also observed; the dairy industry in Zimbabwe was dominated by large-scale production, which contributes 98% of the national supply, while small-scale dairy systems contribute 2% [19]. In Argentina, from 1988 to 2010, small-scale dairy production units in the provinces of Buenos Aires were reduced by 80% [20]. This tells us that this trend has occurred in developing countries of different geographical areas.

Cervantes and Álvarez [21] and Camacho-Vera et al. [15] mentioned that the most profitable and competitive SSDPU that have made significant progress in recent decades were those that specialised and implemented a high technological level, that is, incorporated technological innovations and adopted technical principles of the Holstein model, such as Holstein cows, artificial insemination, use of a cooling tank, mechanised milking, use of a tractor and computer control of production processes. It also includes use of improved forages, especially alfalfa and grain-based concentrates, which implied establishing forage crops with irrigation systems, high use of agrochemicals and forage conservation techniques such as silage. Brunett et al. [22] pointed out that SSDPU that incorporate technological innovations are more sustainable than conventional ones; likewise, Espinoza et al. [23] found that producers in the specialised group would be below the poverty line if they depended solely on income from dairy farming. According to this frame of reference, it is hypothesised that the SSDPU found in northeastern Michoacán are the ones with a higher level of technology into their production logic.

The incorporation of technological innovations has been evident in tropical zones and in northern Mexico [15,24]. However, in some regions of the highlands such as Michoacán, as well as in other regions, it seems that specialisation and the adoption of the Holstein model has not been much included as a strategy for SSDPU to survive. It has even been reported that the technological level may not influence the increase in production [25], which leads to the question of whether it is true that in some regions of Mexico, small livestock producers are using a high technological level to survive. To answer this question, the aim of this study is to characterise the SSDPU in the northeastern region of Michoacán, Mexico, to identify the technological level used by producers, in addition to knowing their current socioeconomic status and thus contribute to the knowledge and understanding of this predominant system, providing elements that can help in the planning of public policies oriented to milk production.

## 2. Materials and Methods

### 2.1. Study Area

The research was carried out in the municipality of Maravatío, located in the northeastern region of the state of Michoacán, Mexico (Figure 1), which represents 1.18% of the surface area of the state, and is composed of 136 localities. Its climate is temperate sub-humid with rainfall in summer, average humidity (72.70%) and the temperature range is from 8 to 20 °C. The altitude is between 2000 and 3500 m above sea level and annual rainfall ranges from 700 to 2000 mm [26].

The importance of this municipality lies in its historical agricultural and livestock tradition, which dates to the eighteenth century [12], highlighting the importance of dairy production as an economic activity in the municipality. Currently, 80% of milk production is carried out in SSDPU, and by 2019, Maravatío contributed 2% of the state’s milk supply, which is equivalent to 5,083,624 L [26,28].

### 2.2. Sample Number

The study was performed in the period 2016–2017. To characterise the SSDPU, a typology was carried out using multivariate analysis, which is a tool to understand the diversity of agricultural systems [29].

According to the information provided by the Agricultural Statistical Yearbook 2015 and the Local Livestock Association of Maravatío, Michoacán (Asociación Ganadera Local de Maravatío), a total of 650 farms were identified, which include dairy-producing, meat-producing and dual-purpose farms. A separation was carried out, selecting only dairy farms. This refining of the database showed that there were 465 SSDPUs in the Maravatío municipality. Starting from the total population (465 SSDPU), the formula for finite populations [30] was applied, with which the sample size was obtained:$$n=\frac{\;z^{2}\;\left(p\right)\times \left(q\right)\times \left(N\right)}{\left(N-1\right)\times e^{2}+z^{2}\times \left(p\right)\times \left(q\right)}$$
where *n* is the sample size; *N* is the size of the population or universe; *Z* is the confidence level; *e* is the error; *p* is the probability that the sample is representative; *q* is the probability that the sample is not representative. Considering the total population (*N* = 465), a confidence level of 95%, a standard error of 8% and a probability that the sample represents 50%, the estimated sample size *n* was 114 surveys, which represented 24.52% of the total production unit’s population.

Once the sample number was obtained (114 SSDPU), a stratified random sampling with proportional allocation was performed, having as a stratification factor the number of dairy cattle per herd, which provided the probability of choosing farms with different herd sizes [31]. Table 1 describes in detail the stratified random sampling, where the strata, the range of animals for each stratum, the number of SSDPU that contains that range of animals and the sample for each stratum are shown.

Semi-structured interviews were conducted with dairy farmers, who showed willingness and allowed access to their SSDPU. The interviews included 28 variables, grouped into four sections: (a) technological level: mechanised milking, artificial insemination, training, feeding with concentrate and alfalfa, tractor use, fertilisation with agrochemicals, stabling, access to government programs; (b) socioeconomic aspects of the producer: age, level of schooling, gender (male), experience, family members, main source of income, family labour (FL), hired labour, organisation among producers (OAP), number of marketing channels, multiple activities (diversification with other agricultural and non-agricultural economic activities); (c) production aspects: total inventory of bovines per herd, inventory of cows in production, average production per cow in line (APCL); and (d) variables related to resources: public water network, own land, organic fertiliser, rainfed hectares, irrigated hectares, and drainage. The variables to be included in the interviews were chosen based on relevant factors for characterising livestock production units [32,33].

### 2.3. Statistical Analysis

The SPSS statistical package (version 20) [34] was used to analyse the information. The Shapiro–Wilk test was performed to verify normality and homoscedasticity of variances for quantitative variables and Kruskal–Wallis for qualitative variables (assumptions of normality).

The variables were standardized to obtain the correlation matrix. Once the correlation was performed, those variables that were highly correlated (28 variables) were maintained. The suitability of the sampling was tested using the Kaiser–Meyer–Olkin test, which indicates whether principal component analysis is an appropriate method [35]. In addition, Barlett’s sphericity test was performed, which confirms that the variables are correlated. A matrix of rotated components was developed using Varimax with the Kaiser Normalization method to describe the principal components [36]. Therefore, of the 28 variables used by PCA technique whose eigenvalues greater than one, 12 variables were selected and used for the HCA and Cluster of K-means.

The optimal number of clusters was identified by cluster analysis using Ward’s hierarchical method. The dendrogram indicated the most appropriate number of clusters, according to the experience of the researchers. Once the number of clusters was identified, the non-hierarchical K-means method was used to corroborate the clusters identified by the hierarchical method [37].

Clusters were characterized by having homogeneity with their groups and heterogeneity between groups. For continuous variables, differences between groups were estimated using a one-way ANOVA analysis and compared using Tukey’s test for pairwise comparisons [38]. For categorical variables, the differences between groups were estimated using the Chi-square test [31].

## 3. Results

### 3.1. Principal Component Analysis

In the Kaiser–Meyer–Olkin test, a value of 0.659 was obtained, indicating that the PCA turned out to be an appropriate model [35,39,40]. Bartlett’s sphericity test and the Chi-squared test were significant (*p* < 0.001), confirming the applicability of the PCA [41], resulting in an adequate method of variable reduction that allows choosing those variables that explain an acceptable proportion of the total variance. Table 2 shows the total variance of the four PCs identified, which is equivalent to 69.83% of the variability. These PCs were used for the HCA and Cluster of K-means.

A rotated component matrix was used to obtain a correlation of each variable close to 1 with only one of the factors and close to zero with all the others. This generated an identification of four PCs (Table 3).

### 3.2. Conformation of Factors or Principal Components

PC 1 (33.35%) gave priority to variables related to milk production per cow/day, cattle inventory and hectares used for growing fodder crops. PC 2 (14.30%) corresponds to variables related to the producer, such as age, years in the activity and level of schooling. PC 3 (11.98%) represents family members and livestock indoors. PC 4 (10.19%) corresponds to family and hired labour.

### 3.3. Hierarchical Cluster Analysis

In order to classify the SSDPU by means of an HCA and K-means cluster, the variables of the four PCs were used. In this way, four groups of SSDPU were formed. Table 4, Table 5 and Table 6 show the percentages for categorical variables, mean and standard deviation (SD) for continuous variables.

A summary of each cluster is described below:

Cluster 1 is composed of six SSDPU (5.26%). This cluster contains SSDPU with a high level of productivity (Table 6), having an average production per cow per day of 20.47 L. This cluster includes the largest SSDPU, in terms of herd size, as well as irrigated and rainfed arable land area. This is the cluster with the highest technological level, focused on productive aspects and forage and crop production, mainly alfalfa, corn and oats. Among the technological innovations, they use mechanized milking, tractor, fertilization with agrochemicals, artificial insemination, training and access to government programs (Table 4). Milking is carried out twice a day and the predominant dairy breed phenotypes, in order, are Holstein, Jersey and Montbéliarde. The animals are housed separately in pens according to their physiological stage. Food supplementation is carried out in the pen. While 50% of these producers receive remittances, they are not used in the dairy activity. There is family labour, but in these SSDPU, more external labour is hired. Of the total production per day, 95.27% goes to commercialization and 4.73% goes to family consumption and feeding of lactating calves. This is the cluster with the most marketing channels (Table 5), among which are, in order of importance, sale of milk to the agro-industry, sale to the final consumer of milk and dairy derivatives that are processed and produced in the SSDPU themselves and sale of milk to the intermediary. All of this marketing is carried out locally.

Cluster 2 is composed of 20 SSDPU (17.54%). Its productive level is lower than cluster 1, but higher than those of clusters 3 and 4 (Table 6), having an average production per cow per day of 17.35 ± 3.64 L. In this cluster, there are medium-sized farms in terms of herd size, as well as irrigated and rainfed arable land area (Table 6). The SSDPU have a high technological level, although lower than cluster 1. Among the technological innovations they use are artificial insemination, tractor and fertilization with agrochemicals (Table 4). Milking is carried out twice a day and the predominant dairy breed phenotypes are Holstein, Jersey and Brown Swiss. Some 50% of the herds keep animals in continuous housing, while the other 50% combine housing with grazing. Farmers who combine stabling and grazing supplement with concentrate in the pens. Producers in this cluster have less experience in the activity (Table 5). While 45% of the producers in this cluster receive remittances from abroad, in general, they are not used for dairy farming. They have two main local marketing channels, which are sales to an intermediaries and sales to final consumers, both of milk and dairy products processed within the SSDPU. Of the total production per day, 91.12% is for marketing, while 8.88% is for self-consumption and feeding of calves.

Cluster 3 is the largest, with 66 SSDPU (57.89%). It contains the SSDPU with a low level of productivity, with an average production per cow per day of 13.99 ± 4.67 L. Cluster 3 and cluster 4 include the smallest SSDPU in terms of herd size. This cluster has a lower technological level, especially in aspects related to feeding and access to government programs. The most common technological innovations used include the use of tractors and fertilization with agrochemicals (Table 4). Milking is carried out twice a day and the phenotypes of dairy breeds are mainly Holstein, and to a lesser extent Jersey. Some 27.27% of the SSDPU keep their animals in stables, and the rest combine stabling with grazing or keep them in continuous or seasonal grazing. Feeding may include pastures such as rye grass, as well as concentrates, stubble and agricultural residues. The average level of schooling is primary school, and together with clusters 2 and 4, they integrate more family labour into SSDPU activities (Table 5). The producers that make up this cluster, together with cluster 4, are those who diversify more into other economic activities (Table 6), such as agriculture or salaried work, and have greater experience in the dairy activity, compared to clusters 1 and 2 (Table 5). While 43.9% of the producers receive remittances from a relative who works abroad, mainly the United States of America, they indicated that most of the remittances are used to meet basic needs such as food, clothing and education. Some 86.64% of milk production goes to commercialization in the different local channels, mainly the sale of milk to intermediaries. In this cluster, 84.21% of the production is commercialized and 15.79% is destined for self-consumption and feeding of lactating calves.

Cluster 4 consists of 22 SSDPU (19.30%). It contains SSDPU with a low level of productivity, with an average production per cow per day of 12.23 ± 5.21 L. The SSDPU in this cluster have a low technological level. The technological innovations they use the most are fertilization with agrochemicals, stabling, tractor use and training. Milking is generally carried out twice a day and the predominant phenotype of the animals is Holstein. It is the cluster with the highest use of agrochemical fertilization and the lowest use of organic fertilizers on its crops (Table 6). Some 63.64% of the SSDPU stable their animals and the others combine stabling with grazing or keep them in continuous or seasonal grazing. Feeding can include pastures such as rye grass, as well as concentrates, stubble and agricultural residues. The main marketing channels are sales to intermediaries and sales to the final consumer of both milk and dairy products produced within the SSDPU. Of the production, 86.64% is marketed and 13.36% is used for self-consumption and feeding of lactating calves.

## 4. Discussion

The present study shows that most of the SSDPU found in the municipality of Maravatío, Michoacán, are those that embrace a lower technological level in their production activity. This is reflected in the results obtained, where 77.19% of the SSDPU have a low technological level, while 22.80% have an intermediate to high technological level. Six farms that make up cluster 1 have a high technological level, but they are not representative farms of the average technological level available to the SSDPU of the studied region. This leads to rejecting the hypothesis proposed for this research, since the SSDPU found in the northeastern region of Michoacán are those with the lowest technological level.

Similar results have been described in other developing countries, since several authors who have worked with small-scale agricultural producers indicate that most of the production units have a low technological level or that the most technified SSDPU represent between 12% and 33% of the total analysed [19,42,43,44,45]. These authors focus their research on analysing the technological level of each identified cluster, but they do not put emphasis on describing the factors that determine the survival and prevalence of SSDPU with a low technological level, which highlights the importance of this study.

Although the SSDPU in clusters 1 and 2 (22.80%) have a high technological level and productivity, most of the SSDPU (clusters 3 and 4) have survived through other mechanisms, derived, to a large extent, from the peasant rationale. In this survival, self-consumption and the use of family labour are prioritised [46,47], and economic or accumulation aspects do not necessarily have to be foremost [48]. Due to this rationale, the economic dependence on an activity can be reduced, generating a diversity of food and resources for the support and development of SSDPU [49], in addition to which they are less susceptible to variations in the prices of milk, inputs and raw materials [50].

The economic and production activity of this type of unit is organised around the use of family labour [51]. A close relationship was observed between the less technified SSDPU and the family, with a higher percentage of FL use in clusters 3 and 4 (1.27 ± 0.76 and 1.18 ± 0.8, respectively) in relation to cluster 1 (0.67 ± 0.82) (Table 5). The family labour provides these units with greater resilience in the face of diverse socioeconomic situations, since the use of family labour generally represents a low or null opportunity cost, due to not receiving a fixed salary [52,53].

Another strength of SSDPU with a low technological level is the diversification of activities inside and outside the SSDPU. This characteristic allows them to diversify the families’ livelihood and increase their income, helping to reduce their dependence on a single activity and on factors not controlled by producers [54] and increasing their resilience [48], in addition to the generation of food diversity and resources for the support and development of the SSDPU [49].

In this case, it was found that clusters 3 and 4 do not depend exclusively on the sale of milk and survive to a large extent due to the diversification of other agricultural and non-agricultural activities and income (raising other animal species, trade, salaried work, or services), making them less dependent on variations in the prices of milk, inputs and raw materials, in addition to distributing their time in other activities [50]. However, the sale of milk continues to be essential, as it represents a secure and constant income [12], which provides capital when other income is not available.

In this way, with diversification, higher income is obtained, and this is a strategy to face uncertainty [55,56]. Additionally, as mentioned by Micha et al. [57], SSDPU producers with smaller herds and less degree of intensification in the activity have a greater possibility of dedicating time to other activities, given that dairy activity demands fewer working hours and therefore, makes a balance between their social life and work possible. Something similar happens with producers with smaller herds and a lower degree of intensification and productivity in Maravatío, because they soon end their activity since they tend to have few animals, which gives them guidelines to dedicate time to various economic activities such as agriculture. Although these results apply to the study area, they do not necessarily indicate that having higher technology cannot have flexibility in the activity, since as reported by Simões et al. [58] and Rodenburg [59], systems with high technology such as roboticized reduce the workforce in dairy farms of various sizes, making the milking process more efficient and offering a more flexible lifestyle for producers.

On the other hand, the implementation of technologies by producers depends on the relative costs between the adoption of such tools, the number of animals and the size of the SSDPU, since they could substitute these technologies with the use of family labour when the land size and production are small [53]. However, when production increases, FL will be not enough and producers will look for technological alternatives to face these changes [60,61]; in other words, the implementation of more technologies corresponds to a greater intensification of SSDPU [60], because it is more profitable [53,62] and the investment is justified [52,60].

The above is confirmed by the results of the study, where the clusters that least implemented technologies (clusters 3 and 4), have lower numbers of cattle (13.03 ± 5.52 and 13.45 ± 11.48, respectively) with respect to clusters 1 and 2 (33.16 ± 4.3 and 21.95 ± 6.25, respectively) (Table 6) and, as mentioned above, family labour was also higher in clusters 3 and 4. Consequently, producers in these clusters, having lower production, may also employ higher FL for other agricultural or non-agricultural activities [53] inside or outside the SSDPU.

Some studies focused on the characterisation of animal production systems argue that, as SSDPU adopt a higher technological level, they have more possibilities of being competitive and thereby sustainable [15,22,42], and the higher the productivity of the SSDPU, the better they are able to face the costs and conditions imposed by the market [63]. In the present study, it was evident that the SSDPU that integrate more technological aspects and elements of the Holstein model are the ones with better productive yields, as was the case of cluster 1 (23.06 ± 2.30 APCL). However, this cluster represents only 5.26% of the sample interviewed, i.e., the characteristics of cluster 1 are not those that predominate in the SSDPU in the region, which could indicate that higher production and productivity do not guarantee that they will survive and be more sustainable.

It was observed that market conditions also have an influence on the permanence of the SSDPU, since the SSDPU in cluster 1 were also the ones that presented the greatest number of marketing channels (2.33 ± 1.03). This reflects the difficulty that producers have in marketing a larger volume of milk in the market, where, on average, two buyers are not enough to sell their daily production, cover their costs and remain in business. In contrast, the SSDPU in clusters 2, 3 and 4 require fewer than two channels to market their small volume of production, and although they may be subject to the price paid by the intermediary [7], they have greater security in marketing the total milk produced, reducing the risk of marketing only part of their production, since the purchase and sale of milk with the collectors or intermediaries is based on a relationship of trust and commitment [8,64].

Some authors point out that the OAP is a way to obtain better market conditions and acquire technologies to modernise their production. It is argued that joining efforts through cooperation facilitates negotiation with the dairy industry, which can ensure the purchase and sale of milk at an adequate price [65], since it is more attractive for large companies to collect and purchase a larger volume of milk that can be pooled among producers [12]. Thus, in relation to the OAP, it was found that, regardless of the level of technology and production, the SSDPU seek to organise themselves for the production and marketing of their products in order to survive, but the proportion found in the study was higher for cluster 1 (50%), as they indicated during the interviews that they organise themselves to have better conditions in the market, although this has not been achieved due to the number of marketing channels. 

In the technology transfer programs that promote the Holstein model, emphasis is placed on the OAP to access technical advice, financing and government support to improve the production and infrastructure of the SSDPU. However, the technology transfer model implemented in Mexico does not contain innovations to improve marketing [66]. This stands out in the results for cluster 1; although it combines greater OAP, training, technological incorporation, and production, these have not led them to a safe option in the marketing of their product. This is possibly an answer for most of the SSDPU in the study (cluster 2, 3 and 4), because as long as they do not have the right conditions in the market to offer more milk at a fair price, they will not look for ways to make their SSDPU more productive, and will therefore not incorporate technological innovations. Hence, they do not take risks in the uncertain circumstances prevailing in the market.

In this regard, the dairy industry can be a productive and technological catalyst for SSDPU, as well as a secure market for their production. However, this region has historically had zero influence from the large national and international dairy industries, as producers market their product mainly to the small cheese industry and to intermediaries who sell milk from house to house [8,12]. This non-existent articulation with large industries has meant that in this region of the country, the trend of centralisation and concentration of production is not evident, as in other regions [13,15,67], which has meant the expulsion and segregation of small producers from the activity [20]. As was observed from 1994 to 2004 among producers from the highlands of Jalisco, where between 25% and 41% left the activity [13], or as in Chipilo in Puebla, where 33% left the activity [14]; in both cases, it was due to the high dependence on inputs from abroad and due to high demands on the quality of milk. Some authors point out that the more producers are linked to industrial complexes, the greater the loss of autonomy; where the decisions of large companies predominate, the greater the dependence on foreign inputs and the pressure to adopt technologies that meet their requirements [20,68]. In this respect, being less linked to large industries, the SSDPU of the four clusters have been allowed to survive with a certain autonomy in their operations. 

Furthermore, the results of the characterisation of the SSDPU in the region denote important features to highlight in relation to the productivity of the SSDPU. Although it has been reported that the technological level is related to the level of productivity, other authors mention that the technological level does not necessarily have a positive effect per se on productivity [52], and there are other socioeconomic variables such as the level of schooling, age and experience that influence productivity [25].

It has been argued that the level of schooling of the producer is a variable that is related to a greater possibility of integrating technological innovations and thus increasing yields in the SSDPU [25,69]. Cluster 1 was the cluster that made the most technological innovations; in this cluster, producers with high school level predominate (50%), while in clusters 2, 3 and 4, primary school level predominates (65%, 50% and 50%, respectively). It has been reported that those producers with higher schooling are more willing to make technological changes that could make their unit more productive [70,71], and it is not the case that schooling is itself directly associated with increased production since, according to Camacho-Vera et al. [25], the level of schooling is possibly an intermediary variable that influences other variables such as income, standard of living or investment capacity, which at the same time have an influence on performance. This is notable because cluster 2 has an intermediate to high level of technological incorporation in its SSDPU, with an average schooling level of primary school education, which is commonly associated with low levels of technology. This suggests that a lower level of schooling does not always translate into lower incorporation of technology and productivity in the SSDPU [72].

Something similar happens with age, and some authors mention that there is a relationship between the age of the producer, the adoption of technological innovations and productivity [69]. They indicate that the younger the farmer, the greater the possibilities of integrating technological innovations in production. In this study, cluster 3, which presents a lower technological level, is composed of producers of a lower age (43.59 ± 8.4 years); in contrast, producers who implement more technological innovations (cluster 1), are on average 54 ± 9.4 years old. This has also been observed in another region of Mexico, where the youngest producers (40 years old on average) are those with a low technological level [49]. 

Some authors indicate that the age of the producer correlates positively with experience, i.e., the older the producer, the more experience in the dairy activity. For example, Vilaboa y Díaz [73] mentioned that older and more experienced producers have deep-rooted knowledge and could be more reluctant to introduce technological change. This is exemplified by cluster 4, which comprises producers with greater age and experience in the activity (age = 61.31 ± 9.5 years, experience = 33.27 ± 9.04 years) but a lower technological level and productivity. However, although the producers of cluster 3 have a similar technological level, productivity and experience in the activity (27.74 ± 2.91), they are the youngest (43.59 ± 8.4 years). This may indicate that, more than age, it is the years of experience that influence technological change, along with other variables. 

Thus, the producers who may be more receptive to the incorporation of the Holstein model are those that make up cluster 1, who have less experience in the activity, but have more studies and the largest herds. As mentioned by Camacho et al. [15] and Cervantes and Álvarez [21], as the SSDPU specializes, it tends to concentrate more cattle. This is reflected in the values obtained for the variable Total Bovine Inventory per Herd (33.16 ± 4.30) (Table 5); therefore, they depend to a greater extent on milk production and the need to incorporate technological elements of the Holstein model to increase their productivity and be able to remain in the market.

Therefore, in this region, milk production is an important economic activity, rooted in cultural practices that have been transmitted over the years. The producers who dedicate themselves to this activity, according to the socioeconomic temporality in which they live, are acquiring experience that has been passed down through the years that is reflected in daily practices and in the incorporation or not of technological innovations in order to survive, preferring those that have given them results for years. In this way, they seek to minimise risks rather than maximise profits [51], a principle that shows the peasant rationality of most of the SSDPU in Maravatío. Therefore, the fact that producers have higher production and productivity does not guarantee that they will survive and be more sustainable. 

Finally, this study covered one year of research, yet small-scale production systems are characterised by strong annual variations in the availability of inputs, prices and climatic conditions [74]. Therefore, further work should cover longer periods to determine other effects of these variations and to see if there are other variables that explain the survival of these SSDPU. In addition, sociocultural aspects such as beliefs and social references were not included, so future characterisations should include these aspects, which could provide answers to decisions made by producers in relation to the incorporation of technological innovations [75]. Moreover, the interviews were conducted with the producers, who indicated their vision, but this does not necessarily reflect the perspectives and realities of the other members of the household [76]. Future work should include the points of view or perceptions of the other members of the family in relation to the dairy activity, especially because this region of Michoacán is a high source of migrant labour, which may influence the permanence or withdrawal from the activity. 

## 5. Conclusions

The study identified four statistically different clusters of small-scale milk production units through multivariate analysis techniques in the municipality of Maravatío, Michoacán, where both socioeconomic variables and technical-productive variables were is equally important in forming the clusters. Of the clusters identified, clusters 3 and 4 predominate, representing 77.19% of the total number of SSDPU studied and having a low technological level. Although clusters 1 and 2 have remained in operation through the incorporation of more technology and better production levels in their SSDPU, they represent only 22.80% of the SSDPU studied. This highlights that most of the small-scale dairy units in Maravatío do not require a high technological level to be functional and active, which is contrary to what was mentioned by authors who emphasize that technology is an important factor to survive [15,21,22,23,24]; this means that their continued activity derives to a large extent from the peasant rationality that transcends economic issues and prioritises survival and self-consumption using the strength of the family labour and diversification.

The clusters identified can help different decision-makers to design public policies and differentiated development strategies, according to the needs and resources of each group of producers, prioritising those that are less technified. In addition, these policies and strategies should be discriminated according to the technical-productive and socioeconomic characteristics of each cluster, in order to strengthen the aspects that that characterize them. 

## Figures and Tables

**Figure 1 animals-11-02546-f001:**
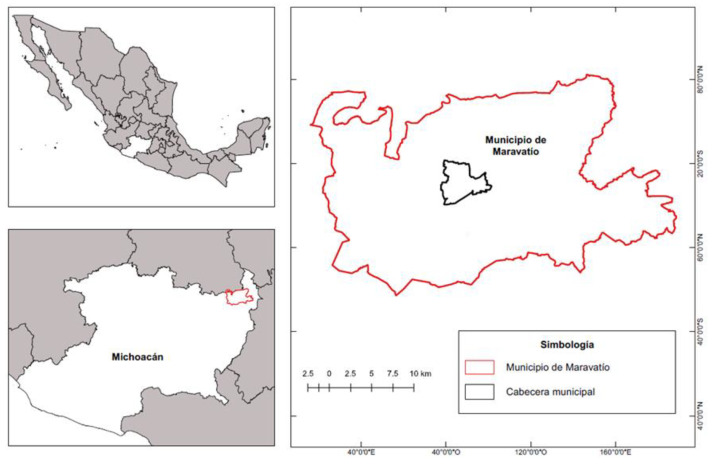
Geographic location of Maravatío, Michoacán, Mexico [27].

**Table 1 animals-11-02546-t001:** Stratification of the SSDPU belonging to the municipality of Maravatío, Michoacán.

Stratum	Ranks of Animals	Number of SSDPU	Sample
1	1–8	67	16
2	9–16	181	45
3	17–24	122	29
4	25–32	65	16
5	33–39	30	8
	TOTAL	465	114

**Table 2 animals-11-02546-t002:** Total variance explained of the variables analysed.

Total Variance Explained									
	Eigenvalues			Sums of Squared Saturations of Extraction			Sum of Squared Saturations of Rotation		
PC	Total	Variance (%)	Cumulative (%)	Total	Variance (%)	Cumulative (%)	Total	Variance (%)	Cumulative (%)
1	4.00	33.35	33.35	4.00	33.35	33.35	3.60	29.97	29.97
2	1.72	14.30	47.65	1.72	14.30	47.65	1.76	14.66	44.62
3	1.44	11.98	59.64	1.44	11.98	59.64	1.67	13.96	58.58
4	1.22	10.19	69.83	1.22	10.19	69.83	1.35	11.25	69.83

**Table 3 animals-11-02546-t003:** Matrix of rotated components of the SSDPU ^a^.

Variable	Component			
	**1**	**2**	**3**	**4**
Average production per cow in line	0.93	−0.026	−0.156	−0.067
Inventory of cows in production	0.862	0.138	−0.098	0.015
Total inventory of bovines per herd	0.85	0.001	0.227	0.034
Irrigated hectares	0.726	0.197	−0.288	−0.131
Rainfed hectares	0.643	0.302	−0.135	−0.143
Age of the farmer	0.217	0.802	−0.115	0.097
Experience	0.076	0.712	−0.256	0.026
Level of schooling	0.027	−0.605	−0.585	0.149
Family members	−0.091	−0.254	0.752	0.071
Stabling	−0.088	−0.087	0.709	0.005
Family labour	0.049	−0.021	0.02	0.924
Hired labour	0.504	−0.143	−0.006	−0.645

Extraction method: Principal Component Analysis. Rotation method: Varimax normalisation with Kaiser. ^a^ The rotation has converged in 5 iterations.

**Table 4 animals-11-02546-t004:** Percentage of implementation of variables related to the use of technological innovations.

Variable	Cluster 1	Cluster 2	Cluster 3	Cluster 4
	(*n* = 6)	(*n* = 20)	(*n* = 66)	(*n* = 22)
Mechanised milking, %	83.33 ^c^	50.00 ^b^	13.63 ^a^	18.18 ^a^
Artificial insemination, %	100.00 ^c^	85.00 ^b^	43.94 ^a^	45.45 ^a^
Training, %	83.33 ^c^	60.00 ^b^	25.76 ^a^	59.10 ^b^
Feeding with concentrate and alfalfa, %	83.33 ^c^	40.00 ^b^	9.09 ^a^	9.09 ^a^
Tractor use, %	100.00 ^c^	85.00 ^b^	72.73 ^a^	63.64 ^a^
Fertilisation with agrochemicals, %	83.33 ^b^	80.00 ^b^	71.21 ^a^	95.45 ^c^
Stabling, %	83.33 ^c^	50.00 ^b^	27.27 ^a^	63.64 ^b^
Access to government programs, %	50.00 ^c^	15.00 ^a^	27.27 ^b^	4.54 ^a^

^abc^ Different letters in the same row mean statistically significant differences (*p* < 0.05). % = Percentage.

**Table 5 animals-11-02546-t005:** Percentage and comparison of means of socioeconomic variables.

Variable	Cluster 1	Cluster 2	Cluster 3	Cluster 4
	(*n* = 6)	(*n* = 20)	(*n* = 66)	(*n* = 22)
	Mean ± SD	Mean ± SD	Mean ± SD	Mean ± SD
Age, *n*	54 ± 9.40 ^b^	48.8 ± 9.40 ^a^	43.59 ± 8.40 ^a^	61.31 ± 9.50 ^c^
Level of schooling				
No schooling, %	0.00 ^a^	10.00 ^b^	4.54 ^a^	18.18 ^c^
Primary %	33.33 ^a^	65.00 ^c^	50.00 ^b^	50.00 ^b^
Secondary, %	16.67 ^b^	5.00 ^a^	24.24 ^c^	9.09 ^a^
High school, %	50.00 ^b^	20.00 ^a^	21.21 ^a^	22.72 ^a^
Gender (male), %	83.33 ^a^	100.00 ^b^	87.87 ^a^	90.90 ^a^
Experience, years	19.83 ± 14.79 ^a^	13.0 ± 7.07 ^a^	27.74 ± 2.91 ^b^	33.27 ± 9.04 ^b^
Family members, *n*	3.83 ± 0.40 ^a^	4.4 ± 1.90 ^a^	5.10 ± 1.50 ^b^	3.95 ± 1.64 ^a^
Main source in income, %	100 ^b^	100 ^b^	72.27 ^a^	86.36 ^a^
Hired labour, *n*	2.33 ± 0.52 ^b^	0.20 ± 0.52 ^a^	0.24 ± 0.43 ^a^	0.27 ± 0.55 ^a^
Family labour, *n*	0.67 ± 0.82 ^a^	1.55 ± 0.76 ^b^	1.27 ± 0.76 ^b^	1.18 ± 0.80 ^b^
Organisation among producers, %	50 ^c^	30 ^b^	13.64 ^a^	40.91 ^b^
Number of marketing channels, *n*	2.33 ± 1.03 ^b^	1.75 ± 0.55 ^a^	1.55 ± 0.56 ^a^	1.73 ± 0.83 ^a^

^abc^ Different letters in the same row mean statistically significant differences (*p* < 0.05); *n*, number. *n* = number, % = percentage.

**Table 6 animals-11-02546-t006:** Percentage and comparison of means of variables related to productivity, land tenure and crop management.

Variable	Cluster 1	Cluster 2	Cluster 3	Cluster 4
	(*n* = 6)	(*n* = 20)	(*n* = 66)	(*n* = 22)
	Media ± SD	Media ± SD	Media ± SD	Media ± SD
Total inventory of bovines per herd, *n*	33.16 ± 4.30 ^c^	21.95 ±6.25 ^b^	13.03 ± 5.52 ^a^	13.45 ± 11.48 ^a^
Inventory of cows in production, *n*	14.83 ± 2.56 ^b^	10.0 ± 2.69 ^b^	4.33 ± 1.99 ^a^	6.36 ± 2.89 ^a^
APCL, L	23.06 ± 2.30 ^c^	17.35 ± 3.64 ^b^	13.99 ± 4.67 ^a^	12.23 ± 5.21 ^a^
Diversification, % *	48.97 ^a^	64.64 ^b^	90.45 ^c^	87.26 ^c^
Drainage, %	100.0 ^b^	85 ^a^	84.85 ^a^	95.45 ^b^
Rainfed hectares, *n*	6.50 ± 1.38 ^b^	3.25 ± 2.05 ^a^	2.41 ± 1.29 ^a^	3.55 ± 2.46 ^a^
Irrigated hectares, *n*	9.17 ± 0.98 ^b^	4.23 ± 1.85 ^a^	2.30 ± 1.71 ^a^	3.73 ± 2.25 ^a^
Organic fertiliser, %	100.00 ^b^	100.00 ^b^	100.00 ^b^	77.27 ^a^
Own land, %	100.00 ^c^	90.00 ^b^	78.79 ^b^	63.64 ^a^
Public water network, %	83.33 ^b^	55.00 ^a^	77.27 ^b^	100.00 ^c^

^abc^ Different letters in the same row mean statistically significant differences (*p* < 0.05); *n*, number; L, litres. * Diversification: percentage of economic activities carried out by producers, taking as 100% the five activities (agriculture, livestock, trade, services sector and salaried work), each activity represents 20%. *n* = number, % = percentage, L = litres.

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
