# Peer review of "Does Small-Scale Livestock Production Use a High Technological Level to Survive? Evidence from Dairy Production in Northeast-ern Michoacán, Mexico"

_animals, 2021, doi:10.3390/ani11092546_

Round 1
Reviewer 1 Report
The article presents general aspects to improve, among them the following stand out:
-The work's hypothesis is poorly stated since it states, "SSDPU that survive in northeastern Michoacán are those that have incorporated a higher level of technology into their production logic," which is not possible since the data does not represent a temporal evolution.
- There are fundamental data that are not present, for example, indicating the breed of cattle, since the comparison of dual-purpose breeds with dairy is not adequate. Thus, the reader does not always know the reality of the breeds used in the study area.
- The units of the variables must be indicated
- Some conclusions are not derived from the results obtained
Particular observations are described below.
Line 62: Indicate if the contribution in volume unit of the SSDPU farms has decreased since the percentage decrease may be due only to the increase in the contribution of the intensive systems.
Line 78. What about feed and nutritional strategies?
Line 133. It is pointed out that a stratified random sampling was performed, but the stratification factor is not indicated
Line 156. The selection methodology of the 12 variables from the 28 chosen variables was not included.
Line 162. Why was the K-mean grouping method used? Was this method selected after evaluating the results of other grouping methods?
Line 168. It is not commented whether normality and heteroscedasticity tests were performed for each variable. Also, which test was used in the case of non-parametric variables.
Line 171. The value of 0.659 for KMO more than adequate indicates that the model is mediocre. Add reference.
Line 187. Was the same PCA used to select the variables? Please clarify. If it is this methodology, it does not seem to be adequate.
Line 187. In addition to the average production of the herd (according to Excel attached), the variable "average production per cow" must be considered in order to comment on the specialization in dairy activity.
Line 202. In tables, use the letter a for the lowest values of each variable.
Line 212. Explain how the diversification percentage was calculated.
Line 236. As noted above, this sentence is not correct since, in order to make this conclusion, two scenarios must be used, before and after the intervention. Since the technological level is not known before the intervention, it is not possible to conclude that there was a decrease.
Line 274. This sentence does not seem entirely accurate, given that technologies, such as milking robots or automatic feeding systems, reduce the time dedicated to the activity
Line 342, It would be convenient to present and comment on some figures
Line 367. Instead, the conclusion is inverse; that is, low education does not imply low technology.
Line 426. Given the herd size of the farms in groups 3 and 4 and the use of dual-purpose breeds, it seems illogical to think that the level of technology affects the continuity of the farms.
Author Response
Consulte el archivo adjunto.

Reviewer 2 Report
The article presents general aspects to improve, among them the following stand out:
The work's hypothesis is poorly stated since it states, "SSDPU that survive in northeastern Michoacán are those that have incorporated a higher level of technology into their production logic," which is not possible since the data does not represent a temporal evolution.
- There are fundamental data that are not present, for example, indicating the breed of cattle, since the comparison of dual-purpose breeds with dairy is not adequate. Thus, the reader does not always know the reality of the breeds used in the study area.
- The units of the variables must be indicated
- Some conclusions are not derived from the results obtained
Particular observations are described below.
Line 62: Indicate if the contribution in volume unit of the SSDPU farms has decreased since the percentage decrease may be due only to the increase in the contribution of the intensive systems.
Line 78. What about feed and nutritional strategies?
Line 133. It is pointed out that a stratified random sampling was performed, but the stratification factor is not indicated
Line 156. The selection methodology of the 12 variables from the 28 chosen variables was not included.
Line 162. Why was the K-mean grouping method used? Was this method selected after evaluating the results of other grouping methods?
Line 168. It is not commented whether normality and heteroscedasticity tests were performed for each variable. Also, which test was used in the case of non-parametric variables.
Line 171. The value of 0.659 for KMO more than adequate indicates that the model is mediocre. Add reference.
Line 187. Was the same PCA used to select the variables? Please clarify. If it is this methodology, it does not seem to be adequate.
Line 187. In addition to the average production of the herd (according to Excel attached), the variable "average production per cow" must be considered in order to comment on the specialization in dairy activity.
Line 202. In tables, use the letter a for the lowest values of each variable.
Line 212. Explain how the diversification percentage was calculated.
Line 236. As noted above, this sentence is not correct since, in order to make this conclusion, two scenarios must be used, before and after the intervention. Since the technological level is not known before the intervention, it is not possible to conclude that there was a decrease.
Line 274. This sentence does not seem entirely accurate, given that technologies, such as milking robots or automatic feeding systems, reduce the time dedicated to the activity
Line 342, It would be convenient to present and comment on some figures
Line 367. Instead, the conclusion is inverse; that is, low education does not imply low technology.
Line 426. Given the herd size of the farms in groups 3 and 4 and the use of dual-purpose breeds, it seems illogical to think that the level of technology affects the continuity of the farms.
Reviewer 3 Report
Reviewer Comments to Authors:
This manuscript is interesting and contributes to work that characterised small dairy producers. However, the following points should be addressed by the authors:
Introduction
Use more similar studies to justify the study, the authors only mention studies in Sweden, Zimbawe and Wisconsin, where 2 of them are in developed countries, that they have a total different production conditions. More detailed examples in Mexico and Latin-American dairy farms production system should be included in the introduction and discussion.
Methodology
Line 136 to 148 and line 152 to 162. Do not use abbreviations in words that are not used at least 3 times in the text.
Line 133 and 134. You have explained how you determined the number of farms used in the experiment but you should explain why there is not a close balance of the number of farms sampled among clusters. Only 6 farms have higher technology level, Are they represent this type of farm in the studied region. Please explain better this point.
Resullts
Table 3. Defined in the footnote FCA
Table 5. Defined in the footnote TIBH, ICP and APCL
Line 214 – 228. Use the word Cluster 1, 2, 3 , 4 rather than Group, as it is managed in the Tables. The information presented in those lines should be more specific for each cluster, because tables only show some specific data (Table 5). Thus, it must show specific data such as: breed of cows, type of technology used, number of milking per day, feed supplementation, among others, to completely characterized the dairy farms.
Discussion
The factors related to the type of farm presented and discussed are interesting. However, it should be emphasised that the commercial objective of the Cluster farms 1 and 2 should be different than 3 and 4, therefore the investment and utilization of technology were different. If there is data, the authors should present the commercial chain of the milk of the 4 types of farms, this is related to the survival of the farm, also.
Line 237. It is said “Similar results have been described in other developing countries, since several authors who have worked with small-scale agricultural producers indicate that more than 90% of the production units have a low technological level [16,35,36], or the most technified SSDPU represent only 33% [37]”. However, other references should be used because it said in other developing countries, while only 2 references are for another countries: Zimbabwe (16) and Colombia (37), but the other references (35 and 36) are studies in Mexico.
Line 244-251. That is true, but the survival of the farms might be related to the aim of the production therefore there are differences in technology. I think, Cluster 1 and 2 have more production because they may use the milk more for a commercial aim thus, they invest in more technology, while Cluster 3 and 4 they may have the aim to produce milk for self-consumption or local commercialization mostly. However, since Cluster characteristic are not clearly described in the results, this conclusion that I have might be incorrect.
The authors should compare their results with similar studies, because they are mentioned in very general form.
Round 2
Reviewer 1 Report
The manuscript has been improved, and the authors have provided appropriate responses to the comments made. However, some minor comments are listed below.
Line 129. It is suggested to include in this section what refers to the selection of dairy farms from the database
Line 176. The explanation of the eigenvectors is confusing; that is, the PCA was carried out with the 28 variables; in this way, the PCs result from an equation that relates the 28 variables. Were the PCs values recalculate for the 12 selected variables and were use these new values for the HCA, or was the PCA performed again using the 12 selected variables and the new PCs were used in the HCA, or were the values of the 12 variables used to perform the HCA ?. Clear out
Concerning this topic, it is suggested to review the following lines to give coherence.
-Line 180: 12 final variables were used for hierarchical cluster analysis.
-Line 202. These PCs with eigenvalues greater than 1 were then used for the subsequent HCA.
Line 196. Indicate the results from the variable selection process
Line 342, The authors mentioned “producers with lower technification and productivity….” I think must be says “producers with smaller herds and / or less degree of intensification in the activity….”
Line 458. I would mention the relationship with the size of the herd, since, although the producers of cluster 1 have less experience, they have more studies and larger herds, in this way the relationship was not unifactorial.
